# GaussianTalker: Speaker-specific Talking Head Synthesis via 3D Gaussian Splatting

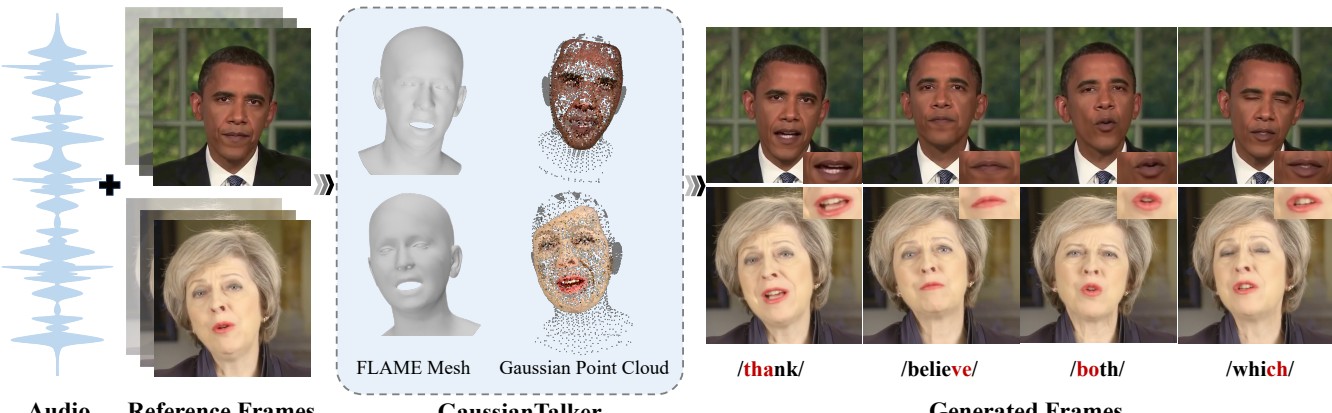

**Figure 1: Given the driving audio and reference frames, GaussianTalker can animate the FLAME mesh of the speaker, which in turn drives the Gaussians associated with it, ultimately synthesizing a high-quality video with precise lip movements.**

## ABSTRACT

Recent works on audio-driven talking head synthesis using Neural Radiance Fields (NeRF) have achieved impressive results. However, due to inadequate pose and expression control caused by NeRF implicit representation, these methods still have some limitations, such as unsynchronized or unnatural lip movements, and visual jitter and artifacts. In this paper, we propose GaussianTalker, a novel method for audio-driven talking head synthesis based on 3D Gaussian Splatting. With the explicit representation property of 3D Gaussians, intuitive control of the facial motion is achieved by binding Gaussians to 3D facial models. GaussianTalker consists of two modules, Speaker-specific Motion Translator and Dynamic Gaussian Renderer. Speaker-specific Motion Translator achieves accurate lip movements specific to the target speaker through universalized audio feature extraction and customized lip motion generation. Dynamic Gaussian Renderer introduces Speaker-specific BlendShapes to enhance facial detail representation via a latent pose, delivering stable and realistic rendered videos. Extensive experimental results suggest that GaussianTalker outperforms existing state-of-the-art methods in talking head synthesis, delivering precise lip synchronization and exceptional visual quality. Our method achieves rendering speeds of 130 FPS on NVIDIA RTX4090 GPU, significantly

exceeding the threshold for real-time rendering performance, and can potentially be deployed on other hardware platforms.

## CCS CONCEPTS

• **Computing methodologies** → **Computer vision**; **Reconstruction**;

## KEYWORDS

talking head synthesis, 3D Gaussian splatting, speaker-specific

## 1 INTRODUCTION

Audio-driven talking head synthesis has attracted significant attention in various fields, including digital avatars, virtual reality, interactive entertainment, and online meetings[32, 44, 47]. This task aims to synthesize a video in which the target speaker's lip movements are in sync with the given audio input.

In existing research, talking head synthesis approaches are primarily categorized into 2D-based and 3D-based approaches. Initially, most 2D-based approaches depended on Generative Adversarial Networks[16] or image-to-image translation[19] to synthesize talking heads in sync with audio. However, the absence of a unified facial model resulted in shortcomings in the identity preservation and pose control of the synthesized videos. More recently, research has been conducted to apply the Neural Radiance Fields (NeRF)[31] to this task. NeRF is a method that models continuous 3D scenes using implicit functions. Compared to 2D-based approaches, NeRF-based approaches can synthesize more lifelike talking head videos by effectively leveraging 3D facial modeling. Nevertheless, these approaches often encounter issues such as unsynchronized or unnatural lip movements, visual jitters, and sporadic artifacts, primarily because NeRF's implicit definition entangles static facial geometry

*MM '24, October 28–November 01, 2024, Melbourne, Australia*
© 2024 Copyright held by the owner/author(s). Publication rights licensed to ACM.
ACM ISBN 978-1-4503-XXXX-X/24/10
https://doi.org/XXXXXXX.XXXXXXX

with dynamic motion, complicating the control over facial poses and expressions.

Recently, 3D Gaussian Splatting (3D GS)[23] has shown impressive achievements in 3D scene reconstruction. 3D GS employs 3D Gaussians as discrete geometric primitives, offering an explicit 3D scene representation and optimizing for real-time rendering performance. In comparison to NeRF, 3D GS not only significantly enhances rendering efficiency and visual quality, but also its paradigm based on 3D Gaussians is easier to control, making it potentially possible to flexibly and intuitively control facial movements. An intuitive strategy is to drive a Gaussian point cloud for facial motion by using parametric 3D facial models[6, 27, 34]. These models typically provide a comprehensive parameter space that allows for the control of facial attributes, such as shape, pose, and expression. By binding the Gaussians to the geometric topology of the model, dynamic talking heads can be generated by synchronizing the displacement of the bound Gaussians with changes in the facial attribute parameters.

Natural lip movements and realistic visual effects are crucial for talking head synthesis. However, the deviation of 3D facial models from a specific speaker's face makes it challenging to synthesize lifelike videos using such a basic binding strategy. Specifically, there are *two main challenges*: **1)** Distribution inconsistencies between the parameters driving the 3D Gaussians and the actual parameters can lead to deviations in point positions. This distribution inconsistency specifically refers to differences in speaking styles. For example, some individuals may speak with a small open mouth, while the driving signal is a wide open mouth. Lip movements that do not align with the speaker's talking style present unnatural effects and even cause lip artifacts. **2)** 3D facial models are limited in modeling complex faces. They can only capture macroscopic muscle movements but not fine details such as wrinkles and teeth. This inherent limitation hinders the 3D Gaussians' accurate representation of a specific face, and the loss of facial details will also lead to visual flicker and artifacts.

For natural lip movements and realistic visual effects, bridging the subtle discrepancies between the 3D facial model and the specific speaker's face is critical. Firstly, audio typically carries speaker identity information. it is necessary to decouple the driving audio from the original speaker's identity information and ensure that the synthesized lip movements closely match the target speaker's style. Secondly, intuitive control of Gaussian attributes helps break through the representation limitations of facial models. With a comprehensive set of blendshapes, these Gaussian attributes can be fine-tuned to capture facial details, refining the facial representation and further fitting a specific speaker.

In this work, we propose GaussianTalker, a framework designed to generate highly natural and realistic talking head videos, adaptable to multiple languages and various timbres of audio input. For dynamic reconstruction, 3D Gaussians are bound to the geometric topology of the parameterized FLAME[27], driving the Gaussians through the deformation of the FLAME mesh, ensuring accurate facial animation. To address the challenge of unnatural lip movements caused by inconsistent distributions, we propose a Speaker-specific Motion Translator. Employing timbre transformation-based contrastive learning for feature decoupling, a speaker-agnostic audio feature is extracted and merged with the target identity embedding

to generate facial poses and expressions that closely align with the target speaker. To confront the challenge posed by unrealistic visual effects caused by the inherent limitations of 3D facial models, the Dynamic Gaussian Renderer introduces Speaker-specific BlendShapes designed to predict the target speaker's latent pose representation and refine facial details through the computation of detail-enhancing Gaussian attributes. It is important to note that our framework is not only capable of achieving speeds well beyond real-time rendering thresholds on contemporary GPUs, but it is also expected to be deployed on other hardware platforms, paving the way for its extensive application across multiple platforms.

The main contributions of this paper are summarized as follows:

- A novel framework for audio-driven talking head synthesis that utilizes 3D Gaussian Splatting bound to FLAME, which generates lifelike rendered videos by associating data from different modalities with specific speakers.
- A Speaker-specific Motion Translator decouples identity and employs personalized embedding for natural lip movements, while a Dynamic Gaussian Renderer refines Gaussian attributes through latent pose to ensure realistic visual effects.
- Extensive quantitative and qualitative experiments show GaussianTalker surpasses state-of-the-art methods in lip-sync and image quality, with its high rendering speed underscoring multi-hardware platform application potential.

## 2 RELATED WORK

### 2.1 Audio-Driven Facial Animation

Most early methods of audio-driven facial animation required establishing a mapping relationship between phonemes and visemes. These methods[3, 4] often overlook individual differences and are overly complex. The emergence of large-scale high-definition video datasets[1, 8, 12, 61], coupled with advancements in deep learning, has propelled the domain of learning-based audio-driven facial animation to the forefront of research. Initial efforts utilized 2D approaches[10, 14, 20, 36, 49, 52, 59, 63, 64], employing Generative Adversarial Networks[16] or image-to-image translation[19], to generate facial animations. However, these approaches fell short of accurately reproducing a speaker's face due to lack of 3D facial modeling, leading to shortcomings in identity preservation and pose controllability. The adoption of 3D Morphable Models (3DMM)[2, 41, 66] has spurred research into audio-driven facial animation using 3DMM. These approaches[22, 30, 46, 51, 58], leveraging 3D modeling, are capable of rendering a more lifelike talking style compared to 2D approaches. Nevertheless, intermediate transformations through 3DMM may lead to information loss[2]. Moreover, prior universal 3D audio-driven animation methods[35, 39] generated vertex predictions with biases toward specific speakers, resulting in animations that fail to express personal styles.

Our method utilizes a parametric 3D facial model to represent the human face, generating facial motion representation from the driving audio. By identity decoupling and personalized embedding, we accommodate the unique styles of different speakers.

### 2.2 Human Face Rendering

Recent studies have adopted Neural Radiance Fields (NeRF)[31] for 3D human head modeling and rendering. NeRF encodes the

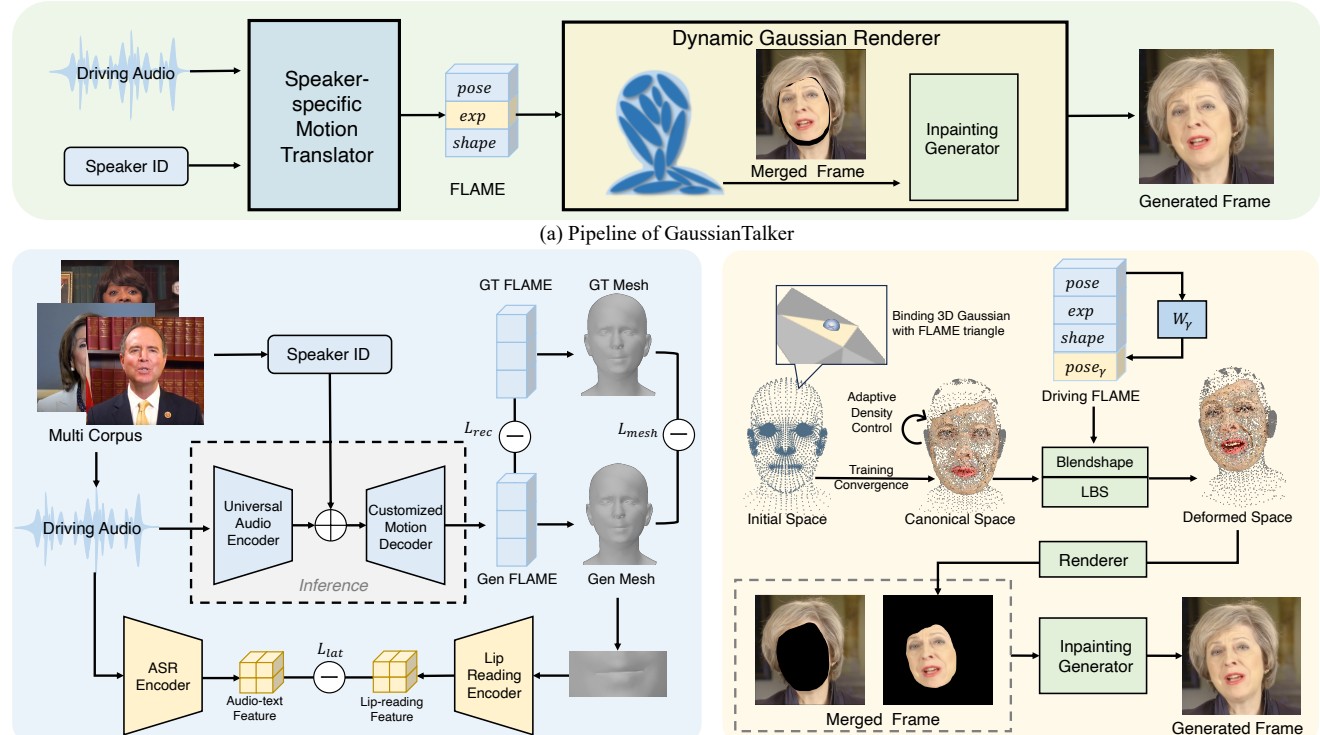

(a) Pipeline of GaussianTalker

(b) Training of Speaker-specific Motion Translator

(c) Training of Dynamic Gaussian Renderer

**Figure 2: Overview of the proposed GaussianTalker. Subfigure (a) depicts speaker-specific FLAME generated from audio, driving Gaussians for rendering. Subfigure (b) illustrates the fusion of speaker-agnostic feature with speaker ID embeddings to decode FLAME. Subfigure (c) shows Gaussians driven by FLAME, subsequently rendering frames.**

scene into a continuous volumetric field using a neural network, which enables high-quality 3D rendering. Following its successful application in dynamic scenes[15, 33, 37], NeRF has been naturally extended to talking head synthesis tasks[7, 17, 26, 29, 43, 45, 55–57]. Some studies have explored the implementation of talking face video generation in an end-to-end way[17, 29, 55]. Subsequent studies have realized several improvements in rendering efficiency[26, 45], few-shot synthesis[28, 43], and lip shape generalization[7, 56, 57].

With the advent of 3D Gaussian Splatting (3D GS)[23], considerable potential has been shown in enhancing both rendering efficiency and quality. Recent work[40, 42, 65] has implemented dynamic head reconstruction based on 3D GS. GaussianHead[50] employs learnable Gaussian diffusion for detailed head generation to accurately reproduce dynamic facial details. MonoGaussian-Avatar[11] utilizes 3D Gaussian representation and deformation fields for learning explicit avatars from monocular portrait videos.

Despite these approaches' effectiveness, audio-driven talking head synthesis faces a significant challenge: cross-modal facial parameter generation from audio fails to precisely capture a specific person's facial nuances. To our knowledge, we first apply 3D GS to this task and overcome the above problem. By Speaker-specific BlendShapes, our method facilitates the synthesis of more lifelike videos, charting a new course in the field of audio-driven high-fidelity talking head synthesis.

## 3 METHOD

### 3.1 Overview

In this section, we provide a detailed description of the architectural components of our proposed framework, GaussianTalker. GaussianTalker employs the parametric FLAME model[27] to serve as a bridge between facial animation and rendering. As shown in Figure 2, the overall framework consists of two main modules: 1) Speaker-Specific Motion Translator, which converts audio signals into speaker-specific FLAME parameters sequence for facial animation control; 2) Dynamic Gaussian Renderer, which utilizes FLAME to drive 3D Gaussians and renders dynamic talking head in real-time. The following subsections detail the design and training mechanisms of each module.

### 3.2 Speaker-specific Motion Translator

Given the diversity of audio inputs, we recognize that model generalization is challenging when solely dependent on video data from specific individuals. Consequently, we train the module using a multilingual, multi-individual dataset to improve its adaptability to diverse audio inputs. However, due to substantial variations in individual speaking styles, the distribution of FLAME parameters generated by the module might diverge from those of the target speaker, potentially compromising the rendered videos' realism. To address this issue, we develop a Universal Audio Encoder that

decouples identity information from content information, and a Customized Motion Decoder that integrates personalization features. Additionally, we refer to SelfTalk[35] and introduce a lip-reading constraint mechanism based on self-supervision to further refine the synchronization of lip movements. This module's detailed architecture and workflow are illustrated in Figure 2(b).

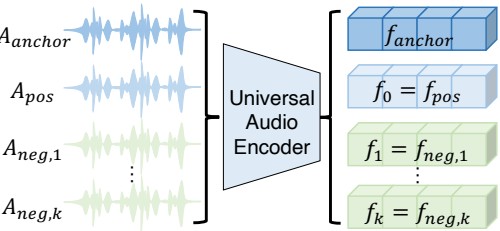

**Figure 3: Negative audios $A_{neg,i}$ are obtained by dividing the audio and positive audio $A_{pos}$ is obtained by timbre conversion. Audios are encoded to get the corresponding features for adversarial learning to fine-tune the encoder to become a Universal Audio Encoder.**

#### 3.2.1 Universal Audio Encoder.
Audio signals often contain both speaker identity information and content information, which are usually tightly coupled. This indicates that in addition to conveying the content of speech, the audio also carries attributes unique to the speaker, such as pitch and timbre. In the task of audio feature extraction, our goal is to capture speech content while excluding information related to speaker identity. Therefore, we start at the audio encoding stage to achieve effective decoupling of identity and content information. We extract audio features using the audio encoder of a pre-trained Wav2Vec 2.0-based multilingual ASR model[54]. As shown in Figure 3, we segment a speaker's audio into $k + 1$ segments, and then randomly select a segment as the anchor audio $A_{anchor} = A_{c1,t1}$. Next, we perform timbre conversion[38] on the anchor audio to obtain positive samples $A_{pos} = A_{c1,t2}$, while the remaining audio segments are used as negative samples. The subscripts $c_i, t_i$ denote the content and timbre of the audio respectively. By encoding the audio, we get the respective feature, whose formulas are as follows:

$$f_t = E_{uni}(A_t; \theta_{uni}), \quad (1)$$

where $f_t$ denotes the audio feature, $E_{uni}$ and $\theta_{uni}$ denote the audio encoder and its learnable parameters, and $A_t$ denotes the audio.

Positive samples are consistent in content with the anchor audio, and we anticipate them to generate similar audio features. For negative sample features, we aim for maximal dissimilarity from the anchor audio feature to achieve de-identification of audio encoding. To achieve this goal, we use the InfoNCE loss[48] for optimization. To maintain the original capability of the encoder, we use $E_{asr}$ and $D_{asr}$, encoder and decoder of the pre-trained Wav2Vec 2.0 model[54] to guide the optimization:

$$\mathcal{L}_{adv} = -\log \frac{\exp(f_{anchor} \cdot f_0 / \tau)}{\sum_{i=1}^{k} \exp(f_{anchor} \cdot f_i / \tau)}, \quad (2)$$

$$\mathcal{L}_{text} = \mathcal{L}_{ctc}(D_{asr}(f_{anchor}), D_{asr}(E_{asr}(A_{anchor}))), \quad (3)$$

where $\tau$ denotes the temperature.

#### 3.2.2 Customized Motion Decoder.
We introduce an identity embedding to generate motions that capture the speaker's talking style, such as the difference in the degree of mouth opening when different speakers talk. The Identity embedding takes personal identity information as input and generates the personalized feature consistent with the audio feature dimensions. Next, the personalized feature is fused with the audio feature to serve as the input of the Customized Motion Decoder. The decoder based on a transformer-based structure, outputs the FLAME parameters sequence directly:

$$\hat{Y}_{1:T} = D_m(E_{id}(p; \theta_{id}) + E_{uni}(A), \theta_m), \quad (4)$$

where $\hat{Y}_{1:T} = (\hat{y}_1, ..., \hat{y}_T)$ denotes the generated FLAME parameters sequence, $D_m$ denotes the motion decoder, $E_{id}$ denotes the identity embedding, $p$ denotes the speaker's identity, and $\theta_{id}, \theta_m$ denote the learnable parameters of the corresponding component, respectively.

#### 3.2.3 Training Object.
The training objectives of Speaker-specific Motion Translator include three components: reconstruction, lip motion smoothness, and potential consistency.

**Reconstruction.** To ensure the generated motion sequence precisely aligns with the ground truth data, we calculate the mean square error of the ground truth $Y_{1:T} = (y_1, ..., y_T)$ and the generated FLAME parameters sequence $\hat{Y}_{1:T}$, as well as the mean square error of the corresponding vertices of the FLAME mesh $V_{1:T} = (v_1, ..., v_T)$ and $\hat{V}_{1:T} = (\hat{v}_1, ..., \hat{v}_T)$:

$$\mathcal{L}_{rec} = \lambda_y \frac{1}{N \times T} \sum_{t=1}^{T} \|y_t - \hat{y}_t\|^2 + \lambda_v \frac{1}{K \times T} \sum_{t=1}^{T} \|v_t - \hat{v}_t\|^2, \quad (5)$$

where $N$ denotes the dimension of FLAME parameters, $K$ denotes the number of FLAME mesh vertices, $T$ denotes the total number of video frames.

**Lip motion smoothness.** To avoid the potential motion jitter problem that results from relying solely on reconstruction loss, we enhance lip motion smoothness by calculating parameter changes across frames:

$$\mathcal{L}_{sth} = \lambda_{sth} \frac{1}{N \times T} \sum_{t=1}^{T} \|(y_t - y_{t-1}) - (\hat{y}_t - \hat{y}_{t-1})\|^2. \quad (6)$$

**Potential consistency.** During training, the generated motion sequence is passed to the lip-reading encoder to extract lip-reading feature. To ensure coherence between the audio-text feature and lip-reading feature, we compute their mean square error:

$$\mathcal{L}_{lat} = \lambda_{lat} \frac{1}{D \times T} \|E_{asr}(A) - E_{lip}(\hat{V}_{1:T}, M_L)\|^2, \quad (7)$$

where $M_L$ denotes the index of lip region, $E_{lip}$ denotes the lip encoder, and $D$ denotes the feature dimension.

The final loss function is formulated as follows:

$$\mathcal{L} = \mathcal{L}_{rec} + \mathcal{L}_{sth} + \mathcal{L}_{lat}. \quad (8)$$

### 3.3 Dynamic Gaussian Renderer
3D GS[23] is a rasterization technique designed for the real-time rendering of photorealistic scenes. It utilizes a collection of 3D Gaussians for modeling, where each Gaussian in the model is characterized by the following learnable parameters: position $u \in \mathbb{R}^3$, indicating the center of the Gaussian; rotation $r \in \mathbb{R}^{3\times3}$, defining its orientation; scaling factor $s \in \mathbb{R}^3$, controlling its size; opacity

$\boldsymbol{\alpha} \in \mathbb{R}$, determining its visibility; and spherical harmonics coefficients $\boldsymbol{\kappa} \in \mathbb{R}^{3 \times (k+1)^2}$, which determine the RGB color $\boldsymbol{c}$ through a k-order spherical harmonic function. This set of parameters for each Gaussian is denoted as $\mathcal{G} = \{\boldsymbol{u}, \boldsymbol{r}, \boldsymbol{s}, \boldsymbol{\alpha}, \boldsymbol{\kappa}\}$. During the rendering process, the color $C$ of each pixel is determined by aggregating the contributions of all overlapping Gaussians, as described by the following equation:

$$C = \sum_{i \in N} \boldsymbol{c}_i \boldsymbol{\alpha}_i \prod_{j=1}^{i-1} (1 - \boldsymbol{\alpha}_j). \tag{9}$$

We aim to employ explicit 3D Gaussians to represent facial expressions, but static 3D Gaussians fall short of capturing dynamic changes in expressions. Consequently, we devised a Dynamic Gaussian Renderer that anchors the Gaussians to the FLAME triangles. This approach leverages FLAME's BlendShapes and Skin Weights to control the deformation of the Gaussians. We also incorporate some Speaker-specific Blendshapes to enhance geometric and textural detail in facial rendering and introduce an Inpainting Generator to seamlessly integrate the rendered face with the original image. The complete pipeline is presented in Figure 2(c).

### 3.3.1 Dynamic Deformation.

Given a set of FLAME parameters, $\boldsymbol{\beta}$ for shape, $\boldsymbol{\varepsilon}$ for expression, and $\boldsymbol{\psi}$ for pose changes, the vertices of the FLAME model in the global space, denoted as $\boldsymbol{v}$, are determined by the following motion rules:

$$\boldsymbol{v} = LBS(\boldsymbol{v}_{base} + BS(\{\boldsymbol{\beta}; \boldsymbol{\varepsilon}; \boldsymbol{\psi}\}; W_{bs}), J(\boldsymbol{\psi}), W), \tag{10}$$

where $\boldsymbol{v}_{base}$ represents the vertices of the FLAME mesh in canonical space. $LBS$ signifies the standard Linear Blend Skinning function with weights $W$ and the function $J$ stands for the joint regressor based on the pose $\boldsymbol{\psi}$, both as defined in the FLAME model. The $BS$ operation denotes a linear blend shaping process that creates facial animations by combining blendshapes in accordance with FLAME parameters, weighted by the blendshape weights $W_{bs}$. For a triangle $\mathcal{T}$ of the FLAME mesh, which is defined by its vertices $\{\boldsymbol{v}_0, \boldsymbol{v}_1, \boldsymbol{v}_2\}$ and its edges $\{\boldsymbol{e}_{01} = \boldsymbol{v}_1 - \boldsymbol{v}_0, \boldsymbol{e}_{02} = \boldsymbol{v}_2 - \boldsymbol{v}_0, \boldsymbol{e}_{12} = \boldsymbol{v}_2 - \boldsymbol{v}_1\}$, we establish a local coordinate system based on $\mathcal{T}$. The system's origin, $P$, representing $\mathcal{T}$'s position in global space, is determined by a barycentric combination of the triangle's vertices. The system's orientation in global space is captured by the rotation matrix $R$, and the scale factor $S$ reflects the size of $\mathcal{T}$. These components are delineated as follows:

$$P = \eta_0 \boldsymbol{v}_0 + \eta_1 \boldsymbol{v}_1 + \eta_2 \boldsymbol{v}_2, \tag{11}$$

$$R = [\boldsymbol{n}_0; \boldsymbol{n}_1; \boldsymbol{n}_2], \tag{12}$$

$$S = (\|\boldsymbol{e}_{01}\| + \|\boldsymbol{e}_{02}\| + \|\boldsymbol{e}_{12}\|)/3, \tag{13}$$

where $\boldsymbol{\eta} = (\eta_0, \eta_1, \eta_2)$ is the learnable barycentric coefficients, and the orthogonal unit vectors $[\boldsymbol{n}_0; \boldsymbol{n}_1; \boldsymbol{n}_2]$ are calculated as follows:

$$\boldsymbol{n}_0 = \frac{\boldsymbol{e}_{01}}{\|\boldsymbol{e}_{01}\|}, \boldsymbol{n}_1 = \frac{\boldsymbol{e}_{01} \times \boldsymbol{e}_{02}}{\|\boldsymbol{e}_{01} \times \boldsymbol{e}_{02}\|}, \boldsymbol{n}_2 = \boldsymbol{n}_0 \times \boldsymbol{n}_1. \tag{14}$$

Then a Gaussian is associated within this local space, denoted as $\mathcal{G} = \{\bar{\boldsymbol{u}}, \bar{\boldsymbol{r}}, \bar{\boldsymbol{s}}, \boldsymbol{\alpha}, \boldsymbol{\kappa}, \mathcal{T}, \boldsymbol{\eta}\}$, enabling $\mathcal{G}$ to track the movements of $\mathcal{T}$. Specifically, the attributes $\bar{\boldsymbol{u}}, \bar{\boldsymbol{r}}, \bar{\boldsymbol{s}}$ of $\mathcal{G}$ are defined relative to the local coordinate system, rather than the global space. We initialize its position $\bar{\boldsymbol{u}}$ at the origin, rotation $\bar{\boldsymbol{r}}$ as the identity matrix, and

scale $\bar{\boldsymbol{s}}$ as a unit vector. When rendering, we transform the $\mathcal{G}$ into global space by:

$$\boldsymbol{u} = R \cdot \bar{\boldsymbol{u}} + P, \boldsymbol{r} = R \cdot \bar{\boldsymbol{r}}, \boldsymbol{s} = S\bar{\boldsymbol{s}}, \tag{15}$$

where $\boldsymbol{u}, \boldsymbol{r}, \boldsymbol{s}$ denotes the global position, orientation, scale of $\mathcal{G}$.

During optimization, we employ an adaptive density control strategy similar to the one described by 3D GS[23], to dynamically add or remove Gaussians based on the view space positional gradient and the opacity of each Gaussian. Specifically, when a Gaussian $\mathcal{G}_i$ is anchored to a triangle $\mathcal{T}_i$ of the FLAME mesh, all its derivatives generated by cloning or splitting from $\mathcal{G}_i$, denoted as $(\mathcal{G}_{i,1}, \mathcal{G}_{i,2}, \mathcal{G}_{i,3}, ...)$, inherit the same local coordinate system that we established based on $\mathcal{T}_i$.

### 3.3.2 Speaker-specific BlendShapes.

We have addressed the challenge of capturing expressive facial motion at a coarse level by binding Gaussians to the triangles of the FLAME mesh. However, since FLAME primarily encodes only coarse facial deformations, it remains a challenge to reproduce speaker-specific facial geometry and texture details, such as teeth and wrinkles. To overcome this limitation, we introduce a set of learnable BlendShape (BS) weights that are designed to capture and delineate these nuanced facial features. Initially, we define a Multilayer Perceptron (MLP), designated as $W_\gamma$, which accepts the pose parameters $\boldsymbol{\psi}$ from the FLAME and generates a latent pose representation $\boldsymbol{\gamma}$ as follows:

$$\boldsymbol{\gamma} = W_\gamma(\boldsymbol{\psi}). \tag{16}$$

To refine geometrical details, specialized BS weights, $W_{pos}$ for position and $W_{rot}$ for rotation, are employed to update the corresponding attributes of Gaussian $\mathcal{G}$ within the local space:

$$\bar{\boldsymbol{u}}' = \bar{\boldsymbol{u}} + BS(\boldsymbol{\gamma}; W_{pos}), \bar{\boldsymbol{r}}' = \bar{\boldsymbol{r}} \cdot BS(\boldsymbol{\gamma}; W_{rot}). \tag{17}$$

These revised local attributes, $\bar{\boldsymbol{u}}'$ and $\bar{\boldsymbol{r}}'$, are then converted to global space when rendering, as explained in Equation 15. Similarly, for texture refinement, we apply compensation to $\boldsymbol{\kappa}_0$, the zeroth-order coefficient of $\boldsymbol{\kappa}$ in $\mathcal{G}$, which governs the base color calculated by zeroth-order spherical harmonics function. This adjustment is achieved through BS weights designated as $W_{color}$:

$$\boldsymbol{\kappa}_0' = \boldsymbol{\kappa}_0 + BS(\boldsymbol{\gamma}; W_{color}). \tag{18}$$

With the incorporation of Speaker-specific BlendShapes for facial detailing, $\mathcal{G}$ is represented as $\mathcal{G} = \{\bar{\boldsymbol{u}}', \bar{\boldsymbol{r}}', \bar{\boldsymbol{s}}, \boldsymbol{\alpha}, \boldsymbol{\kappa}_0', \boldsymbol{\kappa}_{rest}, \mathcal{T}, \boldsymbol{\eta}\}$, where $\boldsymbol{\kappa}_{rest}$ denotes the higher-order(1 to k-th) coefficients of $\boldsymbol{\kappa}$.

### 3.3.3 Inpainting Generator.

By incorporating Dynamic Deformation and Speaker-specific BlendShapes, we achieve a consistent alignment of the head pose with the original face image, which enables a stable reintegration of the rendered face into the original video frame. Nonetheless, using an alternative audio source may lead to discrepancies in lip shapes between the rendered and original video frames. These discrepancies are particularly pronounced around the facial contour and chin, resulting in visual inconsistencies. To address these issues and enhance the integrity of the composite image, we introduce an Inpainting Generator, denoted as $F$. It is designed to seamlessly fill in these mismatches and improve the visual continuity of the final frames. The blending process is described by the following equation:

$$I = (1 - M) \cdot I_{ori} + M \cdot F(I_{gau} + (1 - M)I_{ori}), \tag{19}$$

**Table 1: Quantitative results under the self-driven setting. The best and second-best results are in bold and underlined.**

| Methods | PSNR↑ | SSIM↑ | LPIPS↓ | FID↓ | LMD↓ | LSE-C↑ | LSE-D↓ | FPS↑ |
|---|---|---|---|---|---|---|---|---|
| Ground Truth | — | — | — | — | — | 7.465 | 7.131 | — |
| Wav2Lip[36] | 33.6435 | 0.9506 | 0.0602 | 17.44 | 6.048 | **9.129***  | **5.860*** | 23 |
| AD-NeRF[17] | 30.7756 | 0.9200 | 0.1135 | 26.98 | 3.975 | 6.128 | 8.134 | 0.15 |
| RAD-NeRF[45] | 29.9283 | 0.9115 | 0.1075 | 18.73 | 3.519 | 6.096 | 8.147 | 37 |
| ER-NeRF[26] | 29.5774 | 0.9071 | 0.0694 | 13.22 | 3.555 | 6.340 | 7.955 | 38 |
| GeneFace++[56] | 27.4192 | 0.8870 | 0.0920 | 12.69 | 3.942 | 5.920 | 8.378 | 44 |
| Ours | **37.0775** | **0.9676** | **0.0239** | **4.57** | 3.278 | 7.015 | 7.562 | **130** |

where $M$ denotes the facial mask extracted from the original frame by face parsing[62], and $I_{ori}$ and $I_{gau}$ denote the original video frame and the output image of the Gaussian renderer, respectively.

*3.3.4 Training Object.* While training the Dynamic Gaussian Renderer, we focus on three types of optimization objectives: image similarity, Gaussian attributes, and Gaussian semantic category.

**Image Similarity.** The image similarity loss comprises a combination of L1 loss $\mathcal{L}_1$, VGG loss $\mathcal{L}_{vgg}$, and SSIM loss $\mathcal{L}_{ssim}$:

$$\mathcal{L}_{rgb} = \lambda_1 \mathcal{L}_1 + \lambda_2 \mathcal{L}_{vgg} + \lambda_3 \mathcal{L}_{ssim}. \tag{20}$$

**Gaussian Attributes.** We aim to ensure that during optimization, the positions of the Gaussians faithfully align with the associated FLAME triangles and their sizes remain within reasonable bounds to avoid visual jitter and artifacts. To achieve this, we apply constraints on the Gaussians' position and scale:

$$\mathcal{L}_{attr} = \lambda_p \left\| max(0, \bar{u}' - \epsilon_p) \right\|^2 + \lambda_s \left\| max(0, \bar{s} - \epsilon_s) \right\|^2, \tag{21}$$

where $\epsilon_p$ and $\epsilon_s$ serve as predefined thresholds that set the maximum allowable positions for $\bar{u}'$ and scales for $\bar{s}$, respectively.

**Gaussian Semantic Category.** The dynamic deformation aims for the deformed Gaussians to approximate the associated FLAME mesh. Ideally, Gaussians bound to specific triangles, like the lips, should exactly match the corresponding region's color and position in the rendered image. However, it's a challenge to prevent Gaussians from shifting to implausible positions during optimization, resulting in jittering artifacts. To address this issue, we introduce a Gaussian semantic loss $\mathcal{L}_{seg}$. We render an auxiliary semantic segmentation map $M_{aux}$ using the Gaussian Renderer, where instead of utilizing the learned color $c_i$ in Equation 9, we assign a fixed color $c_{fix}$ to each Gaussian point based on its parent triangle's category; for example, assigning red to facial points and yellow to lip points:

$$C_{seg} = \sum_{i=1} c_{fix} \alpha_i \prod_{j=1}^{i-1} (1 - \alpha_j). \tag{22}$$

Then, we compute the mean square error between $M_{gau}$ and $M_{gt}$ obtained by face parsing[62]:

$$\mathcal{L}_{seg} = \lambda_{seg} \| M_{aux} - M_{gt} \|^2. \tag{23}$$

Consequently, the final loss function is formulated as follows:

$$\mathcal{L} = \mathcal{L}_{rgb} + \mathcal{L}_{attr} + \mathcal{L}_{seg}. \tag{24}$$

---

* Wav2Lip is jointly trained with SyncNet, and LSE-C and LSE-D are its optimization objectives. As a result, it obtains better scores than the ground truth.

**Table 2: Quantitative results under the cross-driven setting. The best and second-best results are in bold and underlined.**

| Methods | Testset A | | | Testset B | | |
|---|---|---|---|---|---|---|
| | FID↓ | LSE-C↑ | LSE-D↓ | FID↓ | LSE-C↑ | LSE-D↓ |
| Wav2Lip[36] | 18.34 | **8.380** | **6.985** | 17.18 | **9.106** | **6.479** |
| AD-NeRF[17] | 29.95 | 3.929 | 11.02 | 29.92 | 3.408 | 11.39 |
| RAD-NeRF[45] | 19.55 | 4.028 | 10.78 | 19.18 | 4.035 | 10.93 |
| ER-NeRF[26] | 13.68 | 4.780 | 10.12 | 13.40 | 4.680 | 10.26 |
| GeneFace++[56] | 12.77 | 5.734 | 9.030 | 12.92 | 6.040 | 8.759 |
| Ours | **5.136** | 5.758 | 8.665 | **4.394** | 7.148 | 7.617 |

## 4 EXPERIMENT

### 4.1 Experimental Settings

*4.1.1 Datasets.* To facilitate comparison with other state-of-the-art methods, we utilize the same dataset as AD-NeRF and GeneFace++, comprised of five videos averaging 9085 frames each, at a frame rate of 25 FPS, across English, French, and Korean languages. To improve the Speaker-specific Motion Translator's generalization capabilities, we incorporate data from target speakers, supplemented with 100 samples of Chinese CN-CVS videos[8] and 100 samples of English HDTF videos[61] for joint training. During the data preprocessing phase, we begin by extracting audio and video frames from the source video. Subsequently, facial keypoints are extracted and smoothed[5, 25], then employ EMOCA[13] to obtain FLAME parameters sequence. Finally, we apply the face parsing technique[62], which segments each video frame into distinct facial regions.

*4.1.2 Compared baselines.* We compare our method with several representative methods: 1) Wav2Lip[36]: Sync-expert supervised learning enhances lip sync accuracy; 2) AD-NeRF[17]: NeRF synthesizes audio-driven videos end-to-end; 3) RAD-NeRF[45]: Discrete learnable mesh boosts rendering efficiency; 4) ER-NeRF[26]: Three-plane hash representation enables high-quality rendering; 5) GeneFace++[56]: Pitch-aware audio-to-motion module and landmark LLE method ensure stable rendering.

*4.1.3 Implementation Details.* The official FLAME model lacks vertices and triangles for teeth, so we manually attach it with 262 vertices and 546 faces to depict the teeth and inner mouth regions. These newly added elements are not assigned independent BS and LBS weights but rather are designed to move in sync with the

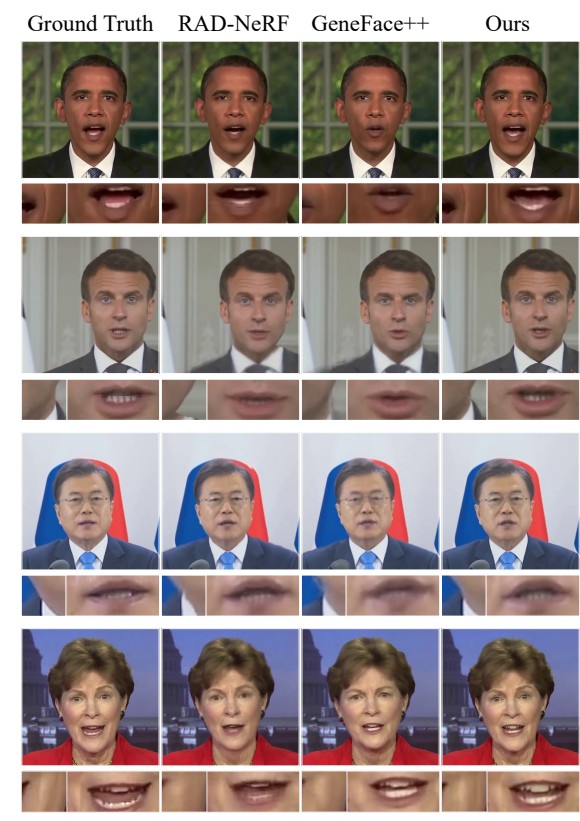

(a) Comparison of six methods on May.

(b) Comparison of latest methods on other speakers.

**Figure 4: The comparison of generated key frame results. We show the ground truth frames for comparing and mark the un-sync and bad rendering quality results with red arrows. Please zoom in for better visualization.**

lip area of the original FLAME model. For GaussianTalker's training, we employ the Adam Optimizer[24] across all modules. The Speaker-specific Motion Translator is trained for 100,000 iterations, with the batch size set to 1. This training takes about 6 hours, using a learning rate of $1 \times 10^{-4}$. Furthermore, we train the Dynamic Gaussian Renderer for 50,000 iterations using a batch size of 8, and this training lasts about 4 hours. While optimizing the Gaussians' attributes, we fine-tune the FLAME parameters within the dataset to ensure they more accurately reflect the face of each frame. All experiments are performed on a single NVIDIA RTX4090 GPU.

## 4.2 Quantitative Evaluation

*4.2.1 Comparison settings.* In quantitative evaluation, we focus on both synthesized quality of the head and accuracy of lip movements. Our comparisons are divided into two settings: 1) Self-driven setting, where each speaker's own audio is used to drive the corresponding reference frame. 2) Cross-driven setting, where we extract two extra audio clips from HDTF[61], named **Testset A** and **Testset B**, to drive each speaker. For each generated result, we rescaled frames into the same size for a fair comparison.

*4.2.2 Evaluation Metrics.* We employ Peak Signal-to-Noise Ratio (PSNR) and Structural Similarity Index Measure (SSIM) for assessing overall image quality. To assess image details, we utilize Learned Perceptual Image Patch Similarity (LPIPS)[60]. To quantify the similarity between the generated video and the ground truth, we employ Fréchet Inception Distance (FID)[18]. For assessing audio-lip synchronization, we utilize Landmark Distance (LMD)[9] to gauge the extent of synchronization with the ground truth, and complementarily evaluate lip movement accuracy through Lip Sync Error Confidence (LSE-C) and Lip Sync Error Distance (LSE-D)[36].

*4.2.3 Evaluation Results.* The results are presented in Table 1. Our results indicate that, in comparison with leading audio-driven talking head video synthesis methods, our method outperforms others, achieving the highest scores across all image quality assessment metrics. Owing to measures implemented to reduce jitter and artifacts, the videos produced by our method more accurately resemble the actual scenes. Regarding lip synchronization, although Wav2Lip excels in the LSE-C and LSE-D metrics owing to its joint training with SyncNet, our method still surpasses others in these metrics. Additionally, our method demonstrates superior performance in the LMD metric, suggesting a closer and more accurate alignment with actual lip movements.

We also compared the generalization ability of each method using FID score and LSE-C and LSE-D metrics with out-of-distribution (OOD) audio inputs, detailed in Table 2. The results show that our

method exhibits outstanding image quality performance, with FID reflecting a considerable enhancement over competing methods. Furthermore, we have significantly improved the model's generalization capabilities by training the Speaker-specific Motion Translator on a multi-individual dataset. This enhancement enables our method to attain superior performance in lip synchronization, as evidenced by the LSE-C and LSE-D.

We also tested the rendering speed. With an NVIDIA RTX4090 GPU and data preloaded into memory, a sequence of video frames can be output at 130 FPS. The output video has the same resolution as the original video. This far exceeds the performance of other methods. In addition, we developed a rendering pipeline for Gaussians with MNN[21] and OpenGL[53] and subsequently deployed GaussianTalker. On devices with the Apple M1 chip, it achieved rendering speeds of 36 FPS, demonstrating its adaptability across various platforms.

### 4.3 Qualitative Evaluation

To more effectively assess image quality and lip synchronization, we present a comparative analysis of our method with others in Figure 4. Compared to our method, Wav2Lip primarily concentrates on lip synchronization yet suffers low image quality, particularly around the mouth area. AD-NeRF and RAD-NeRF cannot naturally perform the blinking process, as evidenced by artifacts appearing in eye closure frames. Moreover, these two methods suffer from head-torso separation when driving cross-identity audio. ER-NeRF demonstrates noticeable jittering in both the head and torso during speech, which severely impacts the realism of the video. GeneFace++ shows limited expressiveness and lip movement variety, failing to perform actions like staring or frowning convincingly.

Most NeRF-based approaches train the head and torso independently, which can compromise the torso's rendering quality in scenes with extensive torso movement. Furthermore, rendering can introduce artifacts stemming from the incomplete separation of the speaker from the background during preprocessing. In contrast, our approach not only delivers superior image clarity but also achieves enhanced lip synchronization performance, particularly with cross-identity and cross-gender audio inputs, showcasing robust generalization capabilities. Please see our supplementary for better visualization.

### 4.4 Ablation study

In this section, we conducted ablation studies in a self-driven setting to validate the effectiveness of each component. Quantitative assessment metrics included PSNR, LPIPS, and LMD. The results are shown in Table 3 and Figure 5.

*4.4.1 Universal Audio Encoder.* We investigate the impact of audio decoupling in the audio encoder, which excludes identity information from the original audio and extracts audio feature that contains only content information. Removing audio decoupling decreases all metrics, with the LMD metric suffering the most, indicating poorer lip synchronization. Figure 5 (a) also illustrates the artifact issues in some of the lip shapes.

*4.4.2 Speaker-specific BlendShapes.* We explore the impact of eliminating the Speaker-specific BlendShapes, which play a crucial role

**Table 3: Ablation study on subject Obama. The best overall results are in bold.**

| Ablation | PSNR↑ | LPIPS↓ | LMD↓ |
|---|---|---|---|
| full | **38.37** | **0.0089** | **3.72** |
| replace Universal Audio Encoder with Wav2Vec 2.0[54] | 36.91 | 0.0104 | 4.55 |
| w/o Speaker-specific BlendShapes | 37.46 | 0.0098 | 3.78 |
| w/o Fine-tune FLAME Parameters | 36.87 | 0.0106 | 4.28 |
| w/o Gaussian Semantic Loss | 37.09 | 0.0104 | 3.97 |

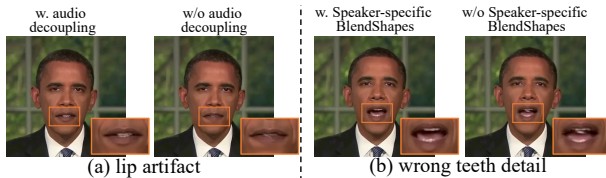

(a) lip artifact      (b) wrong teeth detail

**Figure 5: Ablation study on audio decoupling and Speaker-Specific BlendShape. Removing them will lead to (a) and (b).**

in depicting facial details. The absence may cause the Gaussian renderer to generate artifacts in regions that are not explicitly governed by FLAME parameters, like teeth and wrinkles, resulting in distortions, particularly in fine features, as shown in Figure 5 (b).

*4.4.3 Fine-tune FLAME Parameters.* This optimization aligns the FLAME parameters with the original video and reduces inter-frame jitter. Without it, the Gaussian point cloud learns biased facial representations. As a result, the visual jitter issue becomes more pronounced and the image quality metrics are significantly declined.

*4.4.4 Gaussian Semantic Loss.* We investigate the effect of ablating Gaussian semantic loss, which clarifies the binding relationship between 3D Gaussians and FLAME and further normalizes the motion of Gaussians. Without the Gaussian semantic loss, the speaker will jitter slightly, leading to a reduction in image quality metrics.

## 5 CONCLUSION

In this work, we propose GaussianTalker, a novel framework for audio-driven talking head synthesis via 3D Gaussian Splatting integrated with the FLAME model. GaussianTalker associates multimodal data with specific speakers, reducing potential identity bias between audio, 3D mesh, and video. The Speaker-specific FLAME Translator employs identity decoupling and personalized embedding to achieve synchronized and natural lip movements, while the Dynamic Gaussian Renderer refines Gaussian attributes through a latent pose for stable and realistic rendering. Extensive experiments showed that GaussianTalker outperforms state-of-the-art performance in talking head synthesis, while achieving ultra-high rendering speed that significantly surpasses other methods. We believe that this innovative approach will encourage future studies to develop more fluid and lifelike character expressions and movements. By leveraging advanced Gaussian models and generative techniques, the animation of characters will go far beyond mere lip-syncing, capturing a broader range of character dynamics.

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
