# OpenReview forum: "GaussianTalker: Speaker-specific Talking Head Synthesis via 3D Gaussian Splatting"
_acmmm.org/ACMMM/2024/Conference — MM2024 Poster_

### Official Review · Reviewer_3iNa · 2024-05-02

**Rating:** 4
**Confidence:** 3

**Summary:**

The paper introduces a novel method for generating talking head animations driven by audio inputs. This method employs 3D Gaussian Splatting, which significantly improves the accuracy and realism of facial animations, particularly enhancing lip synchronization and visual quality over previous techniques. The system is built around two main components: the Speaker-specific Motion Translator and the Dynamic Gaussian Renderer. The Motion Translator customizes facial movements to match the unique characteristics of the speaker's voice and style, while the Renderer uses speaker-specific blend shapes to add fine details to the face, thus enhancing the fidelity of the generated videos. GaussianTalker achieves impressive performance metrics, notably in rendering speeds suitable for real-time applications, and demonstrates substantial improvements in both quantitative and qualitative assessments over state-of-the-art methods. The study highlights potential applications in virtual reality, digital avatars, and other multimedia platforms, pushing forward the realism and personalization of digital human representations.

**Strengths:**

1. The paper introduces an innovative use of 3D Gaussian Splatting for audio-driven talking head synthesis. This novel approach improves upon the traditional methods by enabling finer control over facial expressions and movements through a parametric facial model (FLAME).
2. GaussianTalker significantly enhances the realism and synchronization of lip movements with the audio input, which are crucial for creating believable talking head videos. Additionally, the method achieves high rendering speeds, making it suitable for real-time applications, a notable improvement over existing state-of-the-art methods.

**Limitations:**

1. Previous NeRF-based methods[1][2] have rendered the entire head (including hair), but this method only renders the facial region, why is it designed this way?
2. The author mentions in line 821 that the inference speed reaches 140FPS after preloading the data into memory, but as far as I know, the inference process of the source code of RAD-NeRF and ER-NeRF includes a data load process, and I'm wondering if that's not a fair comparison in Table 1?
3. The experimental section lacks a user study, which is an important part of the assessment.
4. Experiments need to be compared with additional 2D-based methods to demonstrate validity, such as [3][4].
5. I know it is not compulsory, but publishing the code will greatly benefit the community and ease the difficulty of understanding.

[1] Ye Z, Jiang Z, Ren Y, et al. GeneFace: Generalized and High-Fidelity Audio-Driven 3D Talking Face Synthesis[C]//The Eleventh International Conference on Learning Representations. 2022.
[2] Li J, Zhang J, Bai X, et al. Efficient region-aware neural radiance fields for high-fidelity talking portrait synthesis[C]//Proceedings of the IEEE/CVF International Conference on Computer Vision. 2023: 7568-7578.
[3] Zhang Z, Hu Z, Deng W, et al. Dinet: Deformation inpainting network for realistic face visually dubbing on high resolution video[C]//Proceedings of the AAAI Conference on Artificial Intelligence. 2023, 37(3): 3543-3551.
[4] Zhong W, Fang C, Cai Y, et al. Identity-preserving talking face generation with landmark and appearance priors[C]//Proceedings of the IEEE/CVF Conference on Computer Vision and Pattern Recognition. 2023: 9729-9738.

**Suitability:**

3

---

### Official Review · Reviewer_5NJy · 2024-05-20

**Rating:** 4
**Confidence:** 4

**Summary:**

The paper presents GaussianTalker, a novel framework for audio-driven talking head synthesis that leverages 3D Gaussian Splatting in conjunction with the FLAME model. GaussianTalker decouples identity from motion using a speaker-specific motion translator module to employ personalized embedding for natural lip movements. GaussianTalker achieves real-time rendering performance (130FPS on an Nvidia RTX 4090 GPU) while surpassing state-of-the-art methods in lip-sync and image quality.

**Strengths:**

1. The method proposes to decouple the driving audio from the speaker's identity information, ensuring that the synthesized lip movements closely match the target speaker's style. This enables GaussianTalker to reproduce realistic talking head videos.
2. GaussianTalker achieves real-time rendering speed while capturing fine motion details of the talking face. The method shows superior ability in both self-driven and cross-driven tasks compared with the state-of-the-art methods.
3. The paper detailedly introduces a novel strategy that binds the 3D Gaussian to the FLAME mesh to enable realistic animation. They propose a Gaussian semantic loss $\mathcal{L}_{seg}$ to guarantee the color of each Gaussian conforms to its parent triangle's category.

**Limitations:**

1. The paper assigns a fixed color to each Gaussian point based on its parent triangle's category, while in the original 3DGS[1], the pixel's color is determined by aggregating the contributions of all overlapping Gaussians, which may have different colors. An adequate analysis is necessary to validate the effectiveness and correctness of this modification.
2. While the hyper-parameter settings are mentioned in the supplementary materials, the paper does not include a discussion on how these settings were chosen or their impact on the network’s performance. A detailed examination of the hyper-parameter settings is crucial for understanding and replicating the training process effectively.
3. The method separates the facial part from the torso and the background and renders them independently. Notably, artifacts are observed in the neck area of the generated videos. This raises concerns about potential quality degradation in scenarios with dynamic backgrounds or significant head movements. It would be beneficial for the paper to address these issues and explore potential solutions.

References:

[1] Bernhard Kerbl, Georgios Kopanas, Thomas Leimkühler, and George Drettakis. 2023. 3D Gaussian Splatting for Real-Time Radiance Field Rendering. (2023)

**Suitability:**

3

---

### Official Review · Reviewer_BYRs · 2024-05-22

**Rating:** 4
**Confidence:** 3

**Summary:**

The paper proposes a framework for audio-driven talking head synthesis using 3D Gaussian splatting coupled to FLAME, which generates lifelike rendered videos by associating data from different modalities with specific speakers.

**Strengths:**

1. The paper proposes a framework for audio-driven talking head synthesis using 3D Gaussian splatting coupled to FLAME, which generates lifelike rendered videos by associating data from different modalities with specific speakers.
2. A speaker-specific motion translator decouples identity and uses personalized embedding for natural lip movements, while a dynamic Gaussian renderer refines Gaussian attributes through latent pose to ensure realistic visual effects.

**Limitations:**

1. While the method proposed in this paper effectively addresses the common problem of jitter, it has a significant limitation in that it only generates the central facial region, using ground truth for the remaining parts. In contrast, other methods generate the entire head, which makes the comparison unfair, especially for metrics such as PSNR, SSIM, and LPIPS. In addition, the lip synchronization metrics, LSE-C and LSE-D, do not seem to reach state-of-the-art levels.
2. Since this paper uses a 3D approach and Flame for reconstruction, the authors should provide synthesized results of the head from different views.
3. The innovation in this paper seems to be incremental, leaning towards the use of existing models. Flame and 3DGS are used for rendering. And 3DGS has already been cited in the face domain, as seen in works like GaussianHead and MonoGaussianAvatar [1]. While this is acceptable, the authors should discuss the differences between the 3DGS used for rendering and GaussianHead and MonoGaussianAvatar.
[1] Yufan Chen, Lizhen Wang, Qijing Li, Hongjiang Xiao, Shengping Zhang, Hongxun Yao, and Yebin Liu. 2023. Monogaussianavatar: Monocular gaussian point-based head avatar. arXiv preprint arXiv:2312.04558 (2023).

**Suitability:**

2

---

### Official Review · Reviewer_QNp1 · 2024-05-23

**Rating:** 4
**Confidence:** 3

**Summary:**

This paper presents a talking head method based on dynamic 3DGS. The method introduces a speaker-specific motion translator that can generate precise facial motion parameters consistent with the target speaker's style based on the input audio. Moreover, the authors propose a dynamic Gaussian renderer that combines FLAME with 3D Gaussian point rendering, achieving realistic facial rendering and high frame rate real-time generation.

**Strengths:**

1. This papaer realizes dynamic Gaussians by binding FLAME with 3D Gaussian Splatting, achieving real-time driving effects at 130 fps, surpassing previous methods based on NeRF，high FPS real-time rendering has certain application value..
2.The Speaker-specific Motion Translator presented in this work offers a fresh perspective by generating driving coefficients tailored to the target speaker.
3.The proposed method surpasses the compared methods in lip synchronization, image quality, and rendering speed.

**Limitations:**

1.Is this paper the first talking head method based on 3DGS? If not, please provide a comparison with other 3DGS-based methods.
2.The proposed method segments the face and other head regions, only driving the facial area. It does not enable control over head pose. Although an Inpainting Generator is used to stitch the face, head, and background together, it may still introduce certain distortions.

**Suitability:**

2

---

### Meta-Review · Area_Chair_ZECg · 2024-06-30

**Recommendation:** Accept (Poster)
**Confidence:** 4

**Metareview:**

The paper described a talking head video generation method which combines gaussian splatting with a parametric 3DMM. The Reviewers seem to have all a positive feeling about this paper, with one reviewer that changed mind after rebuttal and suggest weak rejection.

The paper seems to have several positive aspects, the most important being:
- Combining gaussian splatting with a parametric 3DMM is deemed novel and providing a "fresh perspective" for adding controllability.
- Results are good, surpassing the state of the art both in terms of quality and generation speed.
- Lots of potential practical applications.

On the other hand, some concerns are raised. Most of them were clarified in the rebuttal. The most critical issues are:
- The method only renders the facial region, while other frame details are taken from the ground-truth. This leads to artifacts, mostly in the neck region.
- Given the above, some metrics could be biased.

Overall, the AC believes that the paper is valuable, and  the rebuttal was clear in solving reviewers' concerns. It provided additional evidence in support. Also, the rationale behind the choice of rendering only the facial part is reasonable i.e. audio features are totally uncorrelated to, for example, hair or background. Using those to generate such visual details is counterproductive. This seems a clever observation.

While there are diverging opinions (3 BA and 1 WR), the AC thinks positive aspects overcome negative ones. There still might be an issue with the metrics as their are computed: the AC agrees with BYRs that computing SSIM, LPIPS and FID in this setting is not fair with respect to previous methods that generate larger part of the images, yet it also shows that the proposed solution works better than rendering the entire head, given the reasoning above. The lip-sync metric instead is robust to that and supports the paper claim.

Ultimately, the AC thinks that the paper is worth acceptance, given that such experimental detail is specified in the camera ready and perhaps made also clear in the tables.